



# Evaluation of NU-WRF Performance on Air Quality Simulation under Various Model Resolutions – An Investigation within Framework of MICS-Asia Phase III

Zhining Tao[1,2], Mian Chin[2], Meng Gao[3], Tom Kucsera[1,2], Dongchul Kim[1,2], Huisheng Bian[2,4], Jun-ichi Kurokawa[5], Yuesi Wang[6], Gregory R. Carmichael[7], Zifa Wang[6,8,9], and Hajime Akimoto[10]

1. Universities Space Research Association, Columbia, MD, USA

2. NASA Goddard Space Flight Center, Greenbelt, MD, USA

3. John A. Paulson School of Engineering and Applied Sciences, Harvard University, Cambridge, MA, USA

4. University of Maryland at Baltimore County, Baltimore, MD, USA

5. Japan Environmental Sanitation Center, Asia Center for Air Pollution Research, Niigata, 950-2144, Japan

6. State Key Laboratory of Atmospheric Boundary Layer Physics and Atmospheric Chemistry, Institute of Atmospheric Physics, Chinese Academy of Sciences, Beijing, 100029, China

7. Center for Global and Regional Environmental Research, University of Iowa, Iowa City, IA, USA

8. College of Earth Sciences, University of Chinese Academy of Sciences, Beijing, 100049, China

9. Center for Excellence in Urban Atmospheric Environment, Institute of Urban Environment, Chinese Academy of Sciences, Xiamen, 361021, China

10. National Institute for Environmental Studies, Onogawa, Tsukuba, 305-8506, Japan





## Abstract

Horizontal grid resolution has a profound effect on model performances on meteorology and air quality simulations. In contribution to MICS-Asia Phase III, one of whose goals was to identify and reduce model uncertainty in air quality prediction, this study examined the impact of grid resolution on meteorology and air quality over East Asia, focusing on the North China Plain (NCP) region. NASA Unified Weather Research and Forecasting (NU-WRF) model has been applied with the horizontal resolutions at 45-, 15-, and 5-km. The results revealed that, in comparison with ground observations, no single resolution can yield the best model performance for all variables across all stations. From a regional average perspective (i.e., across all monitoring sites), air temperature modeling was not sensitive to the grid resolution but wind and precipitation simulation showed the opposite. NU-WRF with the 5-km grid simulated the best wind speed, while the 45-km grid yielded the most realistic precipitation as compared to the site observations. For air quality simulations, finer resolution generally led to better comparisons with observations for $O_3$, CO, NOx, and PM2.5. However, the improvement of model performance on air quality was not linear with the resolution increase. The accuracy of modeled surface $O_3$ out of the 15-km grid was greatly improved over the one from the 45-km grid. Further increase of grid resolution, however, showed diminished impact on model performance on $O_3$ prediction. In addition, finer resolution grid showed large advantage to better capture the frequency of high pollution occurrences. This was important for assessment of noncompliance of ambient air quality standards, which was key to air quality planning and management. Balancing the findings and resource limitation, a 15-km grid resolution was suggested for future MICS-Asia air quality modeling activity. This investigation also found out large overestimate of ground-level $O_3$ and underestimate of surface NOx and CO, likely due to missing emissions of NOx and CO.



## 1. Introduction

Air pollution is a threat to human health/climate and detrimental to ecosystem (Anenberg et al., 2010; https://www.who.int/airpollution/ambient/en/). Lelieveld et al. (2015) estimated that over 3 million premature mortality could be attributable to outdoor air pollution worldwide in 2010 based on their analysis of data and the results from a high-resolution global air quality model. Since the turn of $21^{st}$ century, East Asia has undergone remarkable changes in air quality as observed by satellite and ground stations (Jin et al., 2016; Krotkov et al., 2016). In the past decade, haze (fine particle) pollution has become a household name in China and many severe haze events have been reported and their formation mechanisms and associations with global- and meso-scale meteorology have been analyzed (Zhao et al., 2013; Huang et al., 2014; Gao et al., 2016; Cai et al., 2017; Zou et al., 2017). Meanwhile, ground level ozone has been a major air quality concern in China (Wang et al., 2017; Lu et al., 2018), Japan (Akimoto et al., 2015), and South Korea (Seo et al., 2014). In combination with observations from various platforms, chemical transport model (CTM) remains an important tool to understand mechanisms, to investigate spatial-temporal distributions, and to design feasible control strategies of air pollution. However, CTM model uncertainties persist (e.g., Carmichael et al., 2008) and the interpretation of any model results needs caution and exertion of careful analysis.

Inter-model comparison study provides a valuable way to understand model uncertainties and sheds light on model improvements. With this as one of its major goals, the Model Inter-Comparison Study for Asia (MICS-Asia) was initiated in 1998. Since then MICS-Asia has gone through three phases with emphasis on various aspects of air pollution. Phase I focused on long-range transport and deposition of sulfur over East Asia (Carmichael et al., 2002). Phase II expanded the analysis on more pollutants including nitrogen compounds, particulate matter, and ozone, in addition to sulfur (Carmichael et al., 2008). Fast moving to Phase III, MICS-Asia concentrated on three topics with number one aiming at identifying strengths and weaknesses of current air quality models to provide insights on reducing uncertainties (Gao et al., 2018). There are totally 14 CTMs – 13 regional and 1 global – participating in the coordinated model experiment, which simulated air quality over Asia throughout the year 2010. Due to the constrain of computing resources among participating modeling groups, a 45-km horizontal resolution has been commanded for every team to run the year-long experiment.

This relatively coarse spatial resolution raises the question of how representative the model can resolve key issues relevant to air quality and its planning/regulation, e.g., heterogeneous emissions, inhomogeneous land cover and meteorology. For example, Valari and Menut (2008) explored the issue using the CHIMERE model at various horizontal resolutions over Paris. They found out that the ozone simulation was especially sensitive to the resolution of emissions. However, the benefit of increasing emissions resolutions to improve ozone forecast skills was not monotonic and at certain point the forecast accuracy decreased upon further resolution increase. Using the Weather Research and Forecasting Chemistry model (WRF-Chem) with various horizontal resolution (3 ~ 24 km) over the Mexico City, Tie et al. (2010) concluded that a 1 to 6 ratio of grid resolution to city size appeared to be a threshold to improve ozone forecasting skill over mega-city areas: the forecast would be improved significantly when model resolution was below this threshold value. On contrary to Valari and Menut (2008), however, Tie et al. (2010) suggested that the meteorology changes associated with the grid size choice played a more prominent role in contributing to the improvement of ozone forecast skills. More recently, Neal et al. (2017) employed a high-resolution (12 km) air quality model with high-resolution emissions within the Met Office's Unified Model (AQUM) for air quality forecast over the Great Britain.



They found out that AQUM significantly improved the forecast accuracy of primary pollutants
(e.g., $NO_2$ and $SO_2$) but less obviously for secondary pollutants like ozone, as compared with a
regional composition-climate model (RCCM, 50 km horizontal resolution). But there was a
drawback from their conclusion in that the chemical mechanisms and photolysis rates utilized in
AQUM and RCCM were different, complicating the underlying reasons for changes in forecast
skills. Lee et al. (2018) examined the importance of aerosol-cloud-radiation interactions to
precipitation and the model resolution impact of key meteorological processes that affected
precipitation using the Advanced Research WRF model. They found that the coarse model
resolution would lower updraft, alter cloud properties (e.g, mass, condensation, evaporation, and
deposition), and reduce cloud sensitivity to ambient aerosol changes. They further concluded that
the uncertainty associated with resolution was much more than that related to cloud microphysics
parameterization. The resultant meteorological condition change would trigger air quality response
as well.
Despite the progress, the exploration of impacts of model resolution on local air quality
over Asia is rare. Taking advantage of the MICS-Asia platform, we examined the issue over the
MICS-Asia domain using the NASA Unified WRF (NU-WRF, Tao et al., 2013, 2016, 2018;
Peters-Lidard et al., 2015), focusing on the North China Plain (NCP) that was plagued by frequent
heavy air pollution episodes. The investigation would not only assist in gaining insights on how
model horizontal resolution affects simulated meteorology and air quality, but also contribute to
formulation of uncertainties resulted from model resolutions to the MICS-Asia community. The
latter would especially be valuable since most MICS-Asia Phase III model simulations were
conducted at a specific horizontal resolution (i.e., 45-km for most participants).
**2. NU-WRF model and experiment design**
NU-WRF is an integrated regional Earth-system modeling system developed from the
advanced research version of WRF-Chem (Grell et al., 2005), which represents atmospheric
chemistry, aerosol, cloud, precipitation, and land processes at convection-permitting spatial scales
(typically 1-6 km). NU-WRF couples the community WRF-Chem with NASA's Land Information
System (LIS), a software framework including a suite of land surface models (LSMs) that are
driven by satellite/ground observations and reanalysis data (Kumar et al., 2006; Peters-Lidard et
al., 2007). It also couples the Goddard Chemistry Aerosol Radiation and Transport (GOCART)
bulk aerosol scheme (Chin et al., 2002, 2007) with the Goddard radiation (Chou and Suares, 1999)
and microphysics schemes (Tao et al., 2011; Shi et al., 2014) that allows for fully coupled aerosol-
cloud-radiation interaction simulations. In addition, NU-WRF links to the Goddard Satellite Data
Simulator Unit (G-SDSU), which converts simulated atmospheric profiles, e.g, clouds,
precipitation, and aerosols, into radiance or backscatter signals that can directly be compared with
satellite level-1 measurements at a relevant spatial and temporal scale (Matsui et al., 2009, 2013,
2014). In this study, NU-WRF has been employed to carry out the model simulations at various
horizontal resolutions using the same set of physical and chemical configurations.
A nested domain setup was configured to this investigation as shown Figure 1. The 45-km
resolution mother domain (d01) covered the MICS-Asia Phase III study region. The nested 15-km
(d02) and 5-km (d03) domains covered the East Asia and NCP, respectively. This analysis focused
on NCP and its adjacent areas with over 1.1 million square kilometers. The key NU-WRF
configurations included the updated Goddard cumulus ensemble microphysics scheme (Tao et al.,
2011), new Goddard long/shortwave radiation scheme (Chou and Suares, 1999), Monin-Obukhov
surface layer scheme, unified Noah land surface model (Ek et al., 2003) with LIS initialization





(Peters-Lidard et al., 2015), Yonsei University planetary boundary layer scheme (YSU, Hong et
al., 2006), new Grell cumulus scheme off the ensemble cumulus scheme (Grell and Devenyi, 2002)
that allowed subsidence spreading (Lin et al., 2010), 2nd generation regional acid deposition model
(RADM2, Stockwell et al., 1990; Gross and Stockwell, 2003) for trace gases and GOCART for
aerosols. In this investigation, the option of fully coupled GOCART-Goddard microphysics and
radiation schemes (Shi et al., 2014) has been implemented to account for the aerosol-cloud-
radiation interactions.
Anthropogenic emissions were from the mosaic Asian anthropogenic emissions inventory
(MIX, Li et al., 2017) that was developed for the MICS-Asia Phase III. The MIX inventory was
projected to the study domain under the 45-, 15-, and 5-km horizonal resolutions. Fire emissions
were from the Global Fire Emissions Database version 3 (GFEDv3, van der Werf et al., 2010; Mu
et al., 2011) and also projected to the targeted region. Biogenic emissions were computed online
using the Model of Emissions of Gases and Aerosols from Nature version 2 (MEGAN2, Guenther
et al., 2006). Dust and sea salt emissions were also calculated online using the dynamic GOCART
dust emissions scheme (Kim et al, 2017) and sea salt scheme (Gong, 2003), respectively.
The meteorological Lateral Boundary Conditions (LBCs) were derived from the Modern
Era Retrospective-Analysis for Research and Applications (MERRA, Rienecker et al., 2011). The
trace gas LBCs were based on the 6-hour results from the Model for OZone And Related chemical
Tracers (MOZART, Emmons et al., 2010). The aerosol LBCs were from the global GOCART
simulation with a resolution of 1.25 (longitude) by 1 (latitude) degree (Chin et al., 2007). Three
horizontal resolutions varied from 45-km to 5-km with 15-km in between. Terrain-following sixty
vertical levels stretched from surface to 20 hPa with the 1st layer height of approximately 40 meters
from surface. The simulation started on December 20, 2009, and ended on December 31, 2010,
with the first 11 days as the spin-up.
**3. Results**
**3.1. Comparisons with observations**
The NU-WRF results out of different horizontal resolutions have compared with ground
observations using the following statistic measures:
Correlation coefficient: 
$$r = \frac{\sum_{i=1}^{n}(m_i-\overline{m})(o_i-\overline{o})}{\sqrt{\sum_{i=1}^{n}(m_i-\overline{m})^2}\sqrt{\sum_{i=1}^{n}(o_i-\overline{o})^2}}$$

Mean bias: 
$$MB = \frac{1}{n}\sum_{i=1}^{n}(m_i - o_i)$$

Normalized mean bias: 
$$NMB = \frac{\sum_{i=1}^{n}(m_i-o_i)}{\sum_{i=1}^{n}o_i} \times 100\%$$

Root mean square error: 
$$RMSE = \sqrt{\frac{\sum_{i=1}^{n}(m_i-o_i)^2}{n}}$$

Normalized standard deviation: 
$$NSD = \frac{\sqrt{\frac{\sum_{i=1}^{n}(m_i-\overline{m})^2}{n-1}}}{\sqrt{\frac{\sum_{i=1}^{n}(o_i-\overline{o})^2}{n-1}}}$$

Where, $m_i$ and $o_i$ denote for the modeled and observed values at time-space pair $i$; $\overline{m}$ and $\overline{o}$
represent the average modeled and observed values, respectively. $r$ describes the strength and
direction of a linear relationship between two variables – a perfect correlation has a value of 1.
$NMB$ and $MB$ depict the mean deviation of modeled results from the respective observations. A
perfect model simulation yields an $NMB$ and a $MB$ of 0. $RMSE$ measures the absolute accuracy of
a model prediction. The smaller the $RMSE$, the better the model performance is. Similar to $NMB$





and *MB*, a *RMSE* of 0 indicates a perfect model prediction. *NSD* is a measure to check how well
the model can reproduce the variations of observations – a value of 1 represents a perfect
reproduction of observed variations.
### 3.1.1. Meteorology
The 2010 meteorological observations were collected at the standard stations operated by
China Meteorological Administration (CMA, http://data.cma.cn/en). The locations of each site
within our study domain were represented with the black dots in Figure 1. In total there were 77
sites reporting daily average values of wind speed (Wind), air temperature (Temp), and relative
humidity (RH), as well as daily total precipitation (Precip). Figure 2 (top row) shows the Taylor
diagram summarizing $r$, *NMB*, and *NSD* of the comparison of regional mean (average of
observations from 77 sites) daily meteorological variables. Along the azimuthal angle is $r$. *NSD*
is proportional to the radial distance from the origin. *NMB* (sign and range) are represented by
geometric shapes. The statistical measures under 45-, 15-, and 5-km resolutions are represented by
color blue, green, and red, respectively. The closer to the point "Obs" on the Taylor diagram and
smaller of *NMB*, the better the model performance is.
It can be seen that the model horizontal resolution has little impact on surface air
temperature simulation. Regardless of resolution selections, the modeled temperature correlated
very well with the corresponding observations with $r$ values all approaching 0.99. NU-WRF also
reproduced the observed temperature variations well with *NSD* ranging between 1.05 and 1.10.
Meanwhile, *NMB* was within ±1% for all experimented resolutions. *RMSEs* were 1.13 K, 2.26 K,
and 2.02 K for the 45-km, 15-km, and 5-km grids, respectively. The insensitivity of surface air
temperature to the choice of model resolutions was also reported by Gao et al. (2017), who used
WRF to explore the issue for summer seasons at the 36-, 12-, and 4-km resolutions.
On the other hand, the horizontal resolution has a remarkable effect on surface wind speed
as shown in Figure 2 (top row). At 5-km resolution, NU-WRF yielded a $r$ value of 0.75, *NMB* of
approximately 54%, and *NSD* of 1.78. NU-WRF simulated a large variation in wind than the
observed ones. As comparisons, the values of $r$, *NMB*, and *NSD* for 15-km and 45-km were 0.54,
95%, 2.14, and 0.71, 103%, 2.01, respectively. The respective *RMSEs* out of the 45-km, 15-km,
and 5-km grids were 2.87, 2.82, and 1.67 m s⁻¹. It was apparent that 5-km resolution gave the
overall best wind speed simulation compared to the observations, though NU-WRF overestimated
the surface wind speed in all cases. The wind speed overestimate, especially under low wind
conditions, was a common problem in all MICS-Asia participating models and other weather
forecast models (Gao et al., 2018). This overestimate stemmed from many factors, including but
not limited to terrain data uncertainty, poor representation of urban surface effect, horizontal and
vertical grid resolutions, etc. Dr. Yu (2014) in her doctoral dissertation pointed out that surface
wind simulation would be improved upon using more accurate land-use data. This is expected
since surface wind is largely dependent on the land surface characteristics, such as albedo and
roughness. High-resolution grid tends to have more accurate land-use representation seeing the
inhomogeneous nature of land type.
NU-WRF simulations at all three resolutions yielded the similar reproductions of the
observed variations in relative humidity (RH) with the *NSD* ranging between 0.87 and 0.88. The
modeled RH was less variable than the observed one. While the modeled RH at 45-km resolution
($r$ = 0.84) better correlated with the observations than those at the finer resolutions did
(approximately 0.67 for both 15-km and 5-km resolutions), the *NMB* at this resolution was the
largest (-17%) among the three cases. The *NMBs* for 15-km and 5-km cases were -10% and -12%,


respectively. Overall, NU-WRF underestimated the surface RH. The respective *RMSEs* for 45-km,
15-km, and 5-km resolutions were 13.2%, 12.6%, and 13.3%. The simulation with the 15-km grid
appeared to yield the overall best RH in three cases.
It was interesting to find that NU-WRF simulated the best precipitation, as directly
compared to the rain gauge data, when using the 45-km grid. At this resolution, NU-WRF gave $r$
of 0.81, *NMB* of 1.7%, *RMSE* of 3.2 mm day$^{-1}$, and *NSD* of 1.41. As comparisons, the values of $r$,
*NMB*, *RMSE*, and *NSD* for 15-km and 5-km were 0.53, 76%, 5.7 mm day$^{-1}$, 1.71, and 0.52, 80%,
5.8 mm day$^{-1}$,1.72, respectively. Finer resolutions indeed yielded worse results in precipitation
modeling as compared to the site data. This may be because precipitation was a very heterogeneous
phenomenon – finer model grid had larger chances to miss a precipitation event or hit an event
that was not existent, leading to greater overall bias and poorer correlation. On the contrary, Gao
et al. (2017) compared their WRF modeled results to the gridded precipitation based on daily rain
gauge data that were gridded to the 0.125° resolution using the synergraphic mapping algorithm
with topographic adjustment to the monthly precipitation climatology (Maurer et al., 2004). They
reported that the modeled precipitation out of the 4-km resolution was much improved over that
out of the coarser 36- or 12-km resolutions.

### 3.1.2. Air quality

The difference seen in the aforementioned meteorology would cause varied performances
on air quality simulations at various model horizontal resolutions. In this study, the NU-WRF
simulated surface air quality was compared to the corresponding observations. The 2010 ground-
level air quality data were obtained from the Chinese Ecosystem Research Network (CERN,
http://www.cern.ac.cn) operated by the Institute of Atmospheric Physics of Chinese Academy of
Sciences. There were 25 monitoring sites distributed within a 500 km by 500 km area centering
around Beijing, China (open diamond in Figure 1). The site locations and characteristics were
listed in Table 1. 22 out of 25 sites were either in an urban or a suburban setting, with the balance
being in a rural setting. Each site reported hourly concentrations of at least one of the following
six pollutants – ozone ($O_3$), nitrogen oxides (NOx), carbon monoxide (CO), sulfur dioxide ($SO_2$),
and particulate matters with aerodynamic diameters less than 2.5 and 10 μm (PM2.5 and PM10).

### a. Regional average

First, the regional mean (averaged across 25 sites) daily surface concentrations from both
observations and simulations, paired in space and time, were calculated. The $r$, *NME*, and *NSD*
were then computed and illustrated in a Taylor diagram (Figure 2 (bottom row)).
The six pollutants can be put into two groups – one most relevant to ozone photochemistry
including $O_3$, NOx, and CO, and the other closely tied to aerosols including $SO_2$, PM2.5, and
PM10. It was readily seen that the $r$ values of $O_3$, NOx, and CO were not very sensitive to the
choice of model horizontal resolutions. For $O_3$, the $r$ values for 45-km, 15-km, and 5-km grids
were all around 0.85. The respective $r$ values were 0.84, 0.81, 0.80 for NOx, and 0.80, 0.75, 0.73
for CO. In general, however, NU-WRF reproduced the observed variations in $O_3$, NOx, and CO
better with a fine resolution than with a coarse one. *NSD* of 1.23 for $O_3$ at 5-km resolution was the
closest to 1 among three resolutions (1.24 for 15-km and 2.01 for 45-km). *NSDs* were 0.40, 0.36,
0.46 for NOx, and 0.24, 0.27, 0.31 for CO, under the 45-km, 15-km, and 5-km resolutions,
respectively, suggesting that simulations with the finest resolution tended to reproduce the
observed variations better than the ones with coarse resolutions for these three trace gases.
Meanwhile, NU-WRF yielded the smallest bias when employing the fine resolution grid. *NMBs*





for $O_3$ decreased from 115% to 92% when grid resolutions increased from 45-km to 5-km. *NMBs*
were -38%, -30%, -18% for NOx, and -61%, -55%, -51% for CO, under the 45-km, 15-km, and 5-
km resolutions, respectively. It was apparent that NU-WRF overestimated surface $O_3$ but
underestimated NOx and CO, consistent with the findings in the companion MICS-Asia III studies
that based their results on ensemble model simulations (Li et al., 2019; Kong et al., 2019). The
majority of the air quality monitoring sites used in this study were in an urban setting, which
typically were in a VOC-limited regime. This meant that the underestimate of NOx would reduce
the titration that consumed surface $O_3$ leading to its overestimate. We further analyzed the model
bias for daytime (8-18 local standard time) vs. nighttime. It was found that the nighttime biases for
surface $O_3$ and NOx were approximately 2~4 times higher than those of daytime, consistent with
the finding that insufficient NOx titration caused overestimate of modeled surface $O_3$. In the future,
improvement of the emissions inventory accuracy and more realistic temporal emissions
distribution may help improving NU-WRF performance in simulating $O_3$ photochemistry.
NU-WRF simulated less variations in 3 aerosol related pollutants than those of
observations under all applied horizontal resolutions. The *NSDs* ranged from 0.56 (for $SO_2$ at 15-
km resolution) to 0.96 (for PM2.5 at 45-km resolution). Though it reproduced the observed $SO_2$
variations the best (*NSD* = 0.68) with 5-km resolution, NU-WRF yielded the best *NSD* for PM2.5
(0.96) and PM10 (0.92) when 45-km resolution was employed. Similar to 3 trace gases relevant to
surface $O_3$ formation, the choice of model resolution had a limited effect on $r$ statistics. The $r$
values varied from 0.70 (45-km resolution) to 0.76 (both 15- and 5-km) for surface $SO_2$, and from
0.68 (45-km resolution) to 0.63 (5-km) for PM2.5. The $r$ values for PM10 were all around 0.58
under the selected resolutions. The impact of model resolution on *NMBs* showed mixed
information – while the smallest *NMBs* for $SO_2$ (20%) and PM10 (-19%) were achieved using the
45-km resolution, the smallest *NMB* for PM2.5 (1.5%) was observed at the 15-km resolution. The
model underestimate of PM10 was consistent with the findings of the companion investigation
using the multi-model ensemble analysis (Chen et al., 2019).
### 278 *b. Individual site*
The daily average concentrations of each pollutants were calculated and paired in space
and time at each air quality monitoring site. Then the statistics at each individual site was computed.
Figure 3 illustrates the comparisons of *MB*, *RMSE*, and correlation coefficient ($r$) of surface
$O_3$ from different horizontal resolutions at each site. It can be found that there was no single
resolution that yielded the best correlation across all sites. For example, the simulation with the
45-km horizontal resolution gave the best correlation over sites BD, CFD, CZ, HJ, SJZ, SQL, TG,
TJ, TS, XH, XL, YF, YJ, and ZJK. On the other end of spectrum, BJT, DT, and LTH achieved the
best correlation when the 5-km grid was applied. QHD saw the best correlation out of the
simulation with the 15-km resolution. In any cases, however, the variations of $r$ values from
different horizontal resolutions at each site were small (less than 0.04). On the other hand, NU-
WRF yielded the worst *MB* and *RMSE* when employing the 45-km resolution grid, while *MB* and
*RMSE* were similar between simulations with 15-km and 5-km resolutions. Typically, at sites with
urban/suburban settings, *MB (RMSE)* based on the 45-km grid was approximately 15~30%
(20~40%) higher than that out of the 15-km or 5-km grids. It appeared that NU-WRF tended to
have a better performance on ground-level $O_3$ simulation when increasing the horizontal resolution
from 45-km to 15-km, but further finer resolution had diminished impact on improving surface $O_3$
modeling. This was consistent with the finding by Valari and Menut (2008) who concluded that



the benefit of finer horizontal resolution grid to improving surface $O_3$ forecast skill would diminish
at certain point.
Figure 4 shows the PM2.5 case of comparisons of $MB$, $RMSE$, and $r$. Only 10 sites reported
PM2.5 measurements over year 2010. In general, the NU-WRF simulation with the 45-km grid
correlated better to the respective observations than the other 2 resolutions. The only exception
was site BD that saw the best correlation for the 5-km resolution. $MB$ and $RMSE$ results were
mixed with no single resolution giving superior results across all sites. Over 2 rural sites (LS and
XL), the simulations with the 15-km or 5-km grids yielded remarkably smaller $MB$ but correlated
less to the corresponding observations than the one with the 45-km grid. Over 8 urban/suburban
sites, BD, SQL, and TG experienced the smallest $MB$ when employing the 5-km resolution grid,
while TG, TJ, and XH saw the least bias at the 45-km resolution. The smallest $MB$ at BJT and
LTH occurred using the 15-km grid.
At the individual site level, the impact of grid resolution on surface NOx and CO (figures
not shown) modeling was similar to that at the regional average. Finer resolution simulation
generally reduced $MB$ and $RMSE$. The results out of the 45-km grid always had the largest bias.
The underestimates of NOx at least partially explained the overestimate of surface $O_3$ at each site
due to a less efficient NO-titration of $O_3$. This suggested that a higher resolution modeling with
more accurate spatial representation of NOx emissions would help improving its performance on
surface $O_3$ simulations.
The signals for $SO_2$ and PM10 (figures not shown) simulations were mixed as well. For
example, the largest bias for $SO_2$ simulation over sites BD, CZ, GA, HS, LS, QA, QHD, XH, XL,
YF, and YJ occurred when applying the 45-km grid, while the maximum bias over BJT, DT, HJ,
LF, LTH, SJZ, SQL, TG, TJ, TS, ZJK, and ZZ happened at the 5-km resolution. Sites CD and
CFD saw the largest bias at the 15-km resolution. Unlike PM10 that was almost always
underestimated at each site regardless of grid resolutions, $SO_2$ was overestimated at 18 out of 25
sites and underestimated at the remaining 7 sites.
*c. Extreme values*
High concentrations of air pollutants are of more concerns because of their adverse health
effects on both human beings and ecosystem. High pollutant concentrations also pose a greater
risk for non-compliance of the ambient air quality standards. Therefore, evaluations of impacts of
grid resolution on extreme concentrations of air pollutants are desirable.
Figure 5 displays the probability density function distributions of six pollutants based on
hourly surface concentrations across the monitoring sites. This analysis was focused on high
pollutant concentrations with the cutoff values for CO, $O_3$, NOx, $SO_2$, PM2.5, and PM10 being
1.1 ppmv, 60 ppbv, 25 ppbv, 5.5 ppbv, 15 $\mu g\ m^{-3}$, and 30 $\mu g\ m^{-3}$, respectively. It appeared that
NU-WRF, regardless of the grid resolutions, failed to simulate surface CO with concentrations
more than 4 ppmv, likely due to the underestimate of CO emissions (Kong et al., 2019). The grid
resolution appeared to have limited impacts on surface PM10 simulations when its concentrations
were more than 200 $\mu g\ m^{-3}$. On the other hand, the grid resolution showed large impacts on NU-
WRF's capability in simulating high surface concentrations of $O_3$, NOx, $SO_2$, and PM2.5. For
surface $O_3$ with concentrations more than 100 ppbv, the NU-WRF results with the 45-km grid
appeared to better agree with the probability distribution of observations. For surface NOx with
concentrations more than 70 ppbv, the NU-WRF results with the 5-km resolution grid better
mimicked the observed distribution. Modeling with the 5-km grid also yielded the best results of





distributions, in comparisons to the respective observations, of $SO_2$ with concentrations more than
45 ppbv, and of PM2.5 with concentrations greater than 120 µg m$^{-3}$.
Table 2 lists the occurrences of violations of China's national ambient air quality standards
(NAAQS) for the six pollutants from both observations and simulations. It was apparent that NU-
WRF failed to report CO violations at any grid resolutions. No CO NAAQS violation was
simulated but the observation showed that surface CO exceeded the national standard by more
than 1000 times. NU-WRF underestimated the NAAQS exceedances of NOx and $SO_2$. A higher-
resolution grid appeared to be able to catch more violations although the modeled results at the 5-
km resolution only captured 33% and 10% observed exceedances of NOx and $SO_2$, respectively.
NU-WRF overestimated surface $O_3$ and PM2.5 when their concentrations were more than the
corresponding NAAQS. The fine grid resolution (i.e., 5-km) appeared to reduce the overestimation
of surface $O_3$ exceedances largely as compared to the 45-km grid but only marginally compared
with the 15-km grid. Compared to the observed occurrences of surface $O_3$ standard violation
(3,684), the simulated exceedances were 6.7, 2.8, and 2.7 times higher when employing the 45-
km, 15-km, and 5-km resolution grid, respectively. The observations showed 1,343 occurrences
of surface PM2.5 exceedances, while the modeled exceedances were 377, 267, and 231 more for
the 45-km, 15-km, and 5-km grids, respectively. As for surface PM10, the modeled exceedances
were approximately 27%, 43%, and 41% less than the observed one for the 45-km, 15-km, and 5-
km grids, respectively.
**3.2. Inter-resolution comparisons**
It is informative to compare the NU-WRF results out of different horizontal resolutions.
This can help understand the reasons why model resolution matters.
***3.2.1. Emissions***
There were two types of emissions applied in this study. One was the prescribed emissions
out of the anthropogenic and wild fire sources, and the other was emissions computed online using
the real-time meteorology (or dynamic emissions) including emissions from biogenic sources, dust
sources, and sea spray. Amounts and temporal variations of dynamic emissions depended on
surrounding environmental conditions. For example, air temperature and solar radiation regulates
biogenic emissions (Guenther et al., 2006). Surface wind speed plays a major role in both dust
(Ginoux et al., 2001; Chin et al., 2002) and sea salt emissions (Gong, 2003).
For the prescribed emissions, the differences of domain total masses out of each grid were
small (less than 5%). However, the emission gradient around sources of a fine resolution grid
appeared to be sharper than that of a coarse resolution grid. This meant that a coarse grid tended
to distribute the prescribed emissions more evenly into the domain, while a fine grid tended to
produce more extreme concentrations of primary pollutants (emitted directly from a source) such
as NOx and $SO_2$, as shown in Table 2.
Online calculated emissions, on the other hand, displayed large differences in both gradient
and total mass. Similar to the case of prescribed emissions, a fine resolution grid tended to give a
sharper gradient of dynamic emissions than a coarse resolution grid did, as highlighted in Figure
6 ($1^{st}$ row) that illustrated the biogenic isoprene emissions (mol km$^{-2}$ hr$^{-1}$) on a typical summer day.
It was apparent that much more details were simulated using a fine resolution grid - the flow of
Yellow River can even be seen on the 5-km resolution map that was otherwise invisible from the
coarser resolution maps. Meanwhile, the total masses of dynamic emissions showed large
difference out of different resolution grids as listed in Table 3. On an annual basis, the domain




total isoprene emissions were 740,562 tons when estimated using the 45-km grid, approximately
85% and 86% of those with the 15-km and 5-km grids, respectively. The total dust emissions out
of the 45-km grid were 2,431 tons, only 54% and 62% of those based on the respective 15-km and
5-km grids. The percentage contrasts for sea salt emissions were even larger with emissions out of
the 15-km and 5-km grids being 1.3 and 1.6 times more than those of the 45-km grid, respectively.
It should be noted that although they differed greatly between out of the 45-km and 15-km grids,
the dynamic emissions out of the 5-km grid were much closer to those out of the 15-km grid,
partially explaining why the impact of model resolution on surface air quality was less remarkable
by increasing the resolution from 15-km to 5-km than from 45-km to 15-km.
The spatial (gradient) and mass variations in emissions out of different resolution grids
would result in difference in air quality simulations.
### 3.2.2. Meteorology
It's been reported that simulated meteorology varies in response to selections of model grid
resolutions (e.g., Tie et al., 2010; Lee et al., 2018). Meteorology plays an important role in
regulating regional air quality – it affects emissions amount originating from biogenic, dust, and
sea sources; it impacts atmospheric chemical and photochemical transformation; and it directs air
flows and the associated transport of trace gases and aerosols. In this investigation, a few
meteorological parameters key to air pollutant generation and accumulation were analyzed,
including surface wind, air temperature, downward shortwave flux at surface (SWDOWN),
planetary boundary layer height (PBLH), and cloud water (liquid + ice) path (CWP). We focused
on months that were prone to deteriorated PM2.5 (January) and $O_3$ (July) air quality as shown in
Figure 6 and Table 3.
NU-WRF simulated a similar direction of surface wind in July 2010 over the eastern
portion of the domain (2$^{nd}$ row of Figure 6). In general, average wind speed was larger over Bohai
Sea and Yellow Sea than over the surrounding land areas with dominating wind direction being
south and southeast. Based on the results from the 15-km and 5-km grids, the peak average wind
speeds over 4 m s$^{-1}$ were found in Bohai Bay blowing to Tianjin and Beijing. However, such a
peak was absent from the 45-km grid simulation. In the west portion of the domain, the wind
direction changed from southeast in the south to southwest in the north in general. Compared to
the more organized wind directions out of the 45-km grid, wind directions out of the 15- and 5-km
grids were more chaotic. Averaged over the domain, the January mean wind speed out of the 45-
km grid was 2.92 m s$^{-1}$, which were 7% and 16% larger than those of the 15-km and 5-km grids,
respectively. The largest July mean wind speed was again simulated with the 45-km grid, 10% and
12% larger than the corresponding wind speed out of the 15-km and 5-km grids, respectively.
Overall, NU-WRF simulated very similar magnitudes and spatial patterns of surface air
temperature in July (3$^{rd}$ row of Figure 6), regardless of the selections of grid resolutions. Large
portions of the NCP experienced more than 300 K of July average air temperature. The minimum
average temperature of approximately 290 K was found in the central north part of the domain,
which was part of the Mongolian Plateau with the elevation being over 1,500 m above the sea level.
The domain average January and July surface air temperature were around 268 K and 300 K,
respectively, for simulations out of all three grids.
As expected, the modeling results from all three grids (4$^{th}$ row of Figure 6) showed that
July average PBLH over sea was much smaller than that over land. The maximum average PBLH
(more than 1,000 m) was found in the northwest portion of the domain, also in the Mongolian
Plateau with a dominant land cover type of grass. The largest domain-average PBLHs in January



and July were found from the simulations out of the 15-km and 45-km grids, respectively. In
January, the differences of the domain-average PBLHs from different grid resolutions were small
and within 2%. In July, however, such difference can be over 9%.
Regardless of the grid resolutions, NU-WRF simulated a generally southeast-northwest
gradient of SWDOWN in July with the highest flux (over 300 W m$^{-2}$) occurring in the northwestern
domain (5th row of Figure 6). The differences between the maximum and minimum domain
average SWDOWN out of 3 grids were 5.6% and 3.3% in January and July, respectively.
CWP represented the vertical integration of cloud water (including both liquid and ice
phases) contents and can be regarded as a proxy of cloud amount and coverage. Opposite to the
SWDOWN case, NU-WRF modeled a generally northwest-southeast gradient of CWP in July with
the high values found in the southeastern domain (6th row of Figure 6). This was understandable
since cloud reflects and scatters the incoming solar radiation and thus affect SWDOWN. Large
cloud existence tended to reduce the solar flux reaching the underneath Earth surface. The CWP
differences among the model results out of different grid resolutions appeared to be larger than
SWDOWN differences. In July, the domain average CWPs out of the 15-km and 5-km grids were
37% and 33% larger than that of the 45-km grid, respectively. The gaps were even larger in January,
during which the domain average CWPs from the 15-km and 5-km grids were approximately 1.6
times larger than that from the 45-km grid.

### 3.2.3. Air Quality

In response to the aforementioned emissions and meteorological variations resulted from
the selections of model grid resolutions, changes in regional air quality ensued as illustrated in
Figure 7 and Table 3. This figure shows the July average concentrations of ground-level $O_3$ and
its precursors of NOx and CO, as well as the January mean concentrations of surface $SO_2$, PM2.5,
and PM10, during which month the respective pollutants tended to reach high concentrations.
$O_3$ is a secondary pollutant that is formed in the atmosphere through complex
photochemical processes upon existences of its precursors such as NOx and volatile organic
compounds (VOC). Figure 7 (row 1) shows that the spatial distributions of surface $O_3$ are similar
to each other but the concentrations out of the 15-km and 5-km grids are smaller than those from
the 45-km grid. The domain average surface $O_3$ concentration in July was approximately 87 ppbv
based on the results from the 45-km grid, 26% and 25% higher than those out of the 15-km and 5-
km grid, respectively. In January, however, the highest domain average concentration occurred
when the 5-km grid was used, which was 5.3% higher than that out of the 45-km grid.
For the primary pollutants, i.e., NOx, CO, and $SO_2$ (rows 2-4 of Figure 7, respectively),
which were emitted directly by their sources, the spatial distributions of their concentrations
mimicked closely with their emission distributions. High concentrations centered around emission
sources with a reducing gradient outward. The domain average concentrations of these 3 pollutants
out of the 45-km grid results were always the largest in both January and July. The average surface
NOx concentrations from the simulations out of the 15-km and 5-km grids were around 24% lower
than their counterparts out of the 45-km grid in January. In July, the differences were reduced to
7.9% and 11.8% for the 15-km and 5-km grids, respectively. On the other hand, the larger
percentage differences, as compared to the results out of the 45-km grid, occurred in July than in
January for both CO and $SO_2$. For example, the surface CO concentrations out of the 5-km grid
were 12.3% and 30.6% lower than those based on the 45-km grid in January and July, respectively.
The respective ground-level $SO_2$ concentrations from the 5-km grid were 20.5% and 38.9% lower
than those from the 45-km grid in January and July.


It was interesting to note that among the 3 cases, the domain average July surface $O_3$ and
NOx concentrations were both the highest out of the 45-km grid, contrary to the results discussed
in section 3.1.2a where the highest $O_3$ concentration occurred out of the simulation using the 45-
km grid while the highest NOx concentration happened with the 5-km grid. This seemingly
contradicting result was internally consistent. Section 3.1.2a actually depicted the average surface
concentrations in an urban environment (23 of 25 monitoring sites were in an urban/suburban
setting), where surface $O_3$ formation was typically VOC controlled such that NO tended to
consume $O_3$ through titrations. As discussed in section 3.2.1, a 5-km grid gave a much sharper
emissions gradient with anthropogenic emissions concentrating in urban/suburban areas. This led
to higher NOx concentrations around urban/suburban areas out of the simulation with the 5-km
grid, which effectively resulted in lower $O_3$ concentrations there through the NO titration effect.
The domain average discussed in this section, however, was the average covering the vast rural
area that generally was NOx-limited such that surface $O_3$ formation was controlled by the
availability of NOx – more NOx resulting in more $O_3$ through photochemical processes. In this
case, the 45-km grid tended to distribute NOx emissions more evenly in the region, effectively
decreasing the surface NOx concentration in urban areas but increasing it over rural areas. This in
turn increased the domain average surface $O_3$ concentration via photochemistry based on the 45-
km resolution results. In addition, the higher air temperature and stronger SWDOWN in July out
of the 45-km grid as compared to other two resolutions favored more surface $O_3$ generations.
Vertical distributions of $O_3$ tend to have a sizable impact on next day's surface $O_3$ levels
(e.g., Kuang et al., 2011; Caputi et al., 2019). Figure 8 illustrates the domain average profiles of
vertical wind, NOx, $O_3$ (panels a~c), and the average diurnal distribution of surface $O_3$ (panel d)
over July. Here we limited our discussion on the results from the 15- and 5-km grids since 45-km
grid artificially allowed more NOx emissions spreading to rural areas to produce much more $O_3$
as shown in the previous paragraph. Lee et al. (2018) claimed that a coarse resolution model
appeared to lower updraft as compared with a fine resolution modeling. This study agreed with
their finding as illustrated in Figure 8 (panel a). The domain average July vertical wind out of the
simulation with the 5-km grid ranged from 0.25 to 0.45 cm s$^{-1}$ (upward) between 800 hPa and 400
hPa, stronger than the corresponding one out of the 15-km grid. The reason was complex and the
aerosol-cloud interaction induced freezing/evaporation-related invigoration mechanism played a
role (Lee et al., 2018). The stronger upward wind tended to lift more gaseous pollutants up to the
free troposphere as shown in Figure 8 (panel b (NOx) and c ($O_3$)). The pollutants there would have
visible impacts on the following-day surface air quality, especially on $O_3$ levels at night and in the
morning when sun breaks out the nocturnal planetary boundary layer, as evidenced in Figure 8
(panel d). At night with no photochemical formation, surface $O_3$ concentration was largely
controlled by upper-level $O_3$ mixing down, NO titration and $O_3$ dry deposition. With the virtually
same average surface NO concentrations out of the 15- and 5-km grids, the upper-level $O_3$ mixing
down appeared to control the relative magnitudes of surface $O_3$ concentrations simulated using the
15- and 5-km grids. This partially explained why, at night and early morning, the ground level $O_3$
concentrations were higher out of the 5-km grid than from the 15-km grid. During daytime when
the photochemical formation of $O_3$ takes control, the regional average surface $O_3$ concentrations
is largely determined by the availability of $O_3$ precursors (i.e., NOx and VOC) and ambient
environmental conditions. In this case, more spreading NOx emissions out of the 15-km grid
appeared to generate more surface $O_3$ than the 5-km grid did.
PM2.5 and PM10 were mixed pollutants that not only were emitted by various sources but
also were generated in the atmosphere through physical and chemical processes. Figure 7 shows


that high surface concentrations of PM2.5 (more than 120 μg m$^{-3}$, row 5) and of PM10 (more than
170 μg m$^{-3}$, row 6) were still found around the source areas based on the modeling results out of
the 15-km and 5-km grids. However, high PM2.5 and PM10 concentrations spread out to larger
areas based on the results from the 45-km grid as compared to the ones from the finer grid
resolutions. Similar to the primary pollutants, the largest domain average surface concentrations
occurred when a 45-km grid was used for the NU-WRF simulation. The domain average PM2.5
concentrations out of the 15-km and 5-km grids in January were 15.7% and 14% lower than those
from the 45-km grid, respectively. The surface PM2.5 concentration differences among results out
of different grid resolutions grew larger in July, reaching 48% when comparing the result from the
5-km grid to that from the 45-km grid. The domain average surface PM10 concentrations showed
similar pattern to that of PM2.5 with the results out of the 5-km grid being 12.2% and 44.2%
smaller than that from the 45-km grid.
It is worth noting that the magnitudes and spatial distributions of ground-level pollutants
were close to each other between the results out of the 15-km and 5-km grids. This again indicates
that the improvement of fine grid resolution modeling reduces at a certain point. In future MICS-
Asia efforts, a 15-km grid appears to offer the optimized results balanced with performance and
resources.
**4. Summary**
Contributing to MICS-Asia Phase III whose goals included identifying and reducing air
quality modeling uncertainty over the region, this investigation examined the impact of model grid
resolutions on the performances of meteorology and air quality simulation. To achieve this, NU-
WRF was employed to simulate 2010 air quality over the NCP region with three grid resolutions
of 45-km, 15-km, and 5-km. The modeling results were compared to the observations of surface
meteorology archived by CMA, and of ground-level air quality collected in CERN. The inter-
model comparison among the simulation results out of three grids were also conducted to
understand the reasons why model resolution mattered.
The analysis showed that there was no single resolution which would yield the best
reproduction of meteorology and air quality across all monitoring sites. From a regional average
prospective (i.e., across all monitoring sites in this study), the choice of grid resolution appeared
to have a minimum influence on air temperature modeling but affected wind, RH, and precipitation
simulation profoundly. A 5-km grid appeared to give the best wind simulation as compared to the
observations quantified by bias, RMSE, standard deviation, and correlation. Compared to one
using the 45-km grid, the simulated wind speed from a 5-km grid reduced the positive bias by
46.8%. While a 15-km grid yielded the best overall performance on RH modeling, the result out
of the 45-km grid gave the most realistic reproduction of precipitation. The statement on
precipitation should be taken with caution since it was based on the comparison with the site
observations. Seeing the very heterogeneous nature of precipitation, the penalty of model hitting
or missing a rain event was severe. Thus, the coarse grid covering more areas within a grid cell
would reduce chances of mistaken precipitation hitting or missing simulations. However, a
comparison of modeled precipitations to gridded "observation" that was re-constructed using the
synergraphic mapping algorithm with topographic adjustment to the monthly precipitation
climatology showed opposite result, where the fine resolution modeling showed superior
reproduction of precipitation than the coarse resolution simulation (Gao et al., 2017).
The simulated meteorology differences due to the selection of grid resolutions would
consequently lead to differences in air quality simulation. Air pollutant concentrations were



basically determined by their emissions and underlying meteorology that directed their formation (e.g., $O_3$ and aerosols), transport, and removal processes. For the prescribed emissions originated from anthropogenic and wild fire sources, the grid resolution had limited influence on emission amount – less than 5% difference with each other under the different resolution grids - but large impact on emission spatial distribution with sharper emission gradient around sources out of a fine resolution grid than from a coarse resolution one. For the dynamic emissions driven by meteorology, not only was an emission gradient around a source larger out of a higher resolution grid, but also the total emission amount varied greatly. For example, the domain total annual biogenic isoprene emissions from a 5-km grid was about 16% larger than those out of a 45-km grid due to the underlying differences in land cover and meteorology.

Though the impact of grid resolution on air quality varied from location to location, finer grid yielded better results for daily mean surface $O_3$, NOx, CO, and PM2.5 simulations from a regional average perspective. For example, after reducing the grid resolution from 45-km to 15-km, the positive bias of daily mean surface $O_3$ and PM2.5 decreased by 15% and 75%, respectively. Fine resolution modeling was especially beneficial to high pollutant concentration forecast. This was important to air quality management. Taking China's NAAQS as cutoff values for each pollutant, the frequencies of noncompliance occurrences of $O_3$, NOx, $SO_2$, and PM2.5 out of the 5-km grid simulation were much closer to the observations than those out of the 45-km modeling were. It also was worth noting that the benefit of increasing grid resolution to better surface $O_3$ and PM2.5 simulations started to diminish when the horizontal resolution reached 15-km, agreeing with the finding by Valari and Menut (2008).

It should be pointed out that NU-WRF significantly overestimated surface $O_3$ concentration but underestimated ground-level CO and NOx concentrations regardless of grid resolutions. This was true not only on the regional averages but also at majority of the monitoring sites. The missing emissions was believed to be largely responsible for this result (Kong et al., 2019). Underestimate of surface NOx tended to increase ground-level $O_3$ due to the reduced titration effect, especially at night.

In conclusion, grid resolution had a profound effect on NU-WRF performance on meteorology and air quality over the East Asia. Fine resolution grid did not always generate the best modeling results and the proper selection of horizontal resolution hinged on investigation topics for a given set of physics and chemistry choices in a model. With regard to MICS-Asia Phase III whose major goal was to examine air quality, a 15-km horizontal grid appeared to be an appropriate choice to optimize model performance and resource usage.

**Competing interests**

The authors declare that they have no conflict of interest.

**Author contribution**

ZT and MC designed the experiments. ZT, MG, TK, DK, and HB carried out the experiments working on various modeling components. YW collected, organized, and archived the ground air quality measurement data. All authors contributed to model result analysis and interpretation. ZT prepared the manuscript with contributions from all co-authors.

**Acknowledgement**

This work was supported by the NASA's Atmospheric Composition: Modeling and Analysis Program (ACMAP) and Modeling, Analysis, and Prediction (MAP) program. The



authors thank MICS-Asia for its organized platform of discussion and data sharing. This work is
not possible without the supercomputing and mass storage support by NASA Center for Climate
Simulation (NCCS). All data collected and generated for this research are archived and stored on
NCCS servers. Due to the sheer size of data, it is impractical to upload data to a public domain
repository. However, the authors will be happy to share data on an individual request basis.





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



Table 1. Information of Air Quality Observation Sites

| Site Name | Symbol | Longitude | Latitude | Altitude (m) | Setting |
|---|---|---|---|---|---|
| Baoding | BD | 115.441 | 38.824 | 4 | Urban |
| Beijing Tower | BJT | 116.372 | 39.974 | 44 | Urban |
| Chengde | CD | 117.925 | 40.973 | 395 | Urban |
| Caofeidian | CFD | 118.442 | 39.270 | 11 | Urban |
| Cangzhou | CZ | 116.779 | 38.286 | 12 | Urban |
| Datong | DT | 113.389 | 40.089 | 1058 | Urban |
| Gu An | GA | 115.734 | 39.149 | 21 | Rural |
| Hejian | HJ | 116.079 | 38.423 | 66 | Urban |
| Hengshui | HS | 115.656 | 37.742 | 77 | Urban |
| Langfang | LF | 116.689 | 39.549 | 19 | Urban |
| Lingshan | LS | 115.431 | 39.968 | 116 | Rural |
| Longtan Lake | LTH | 116.430 | 39.870 | 31 | Urban |
| Qian An | QA | 118.800 | 40.100 | 54 | Urban |
| Qinhuangdao | QHD | 119.570 | 39.950 | 2.4 | Urban |
| Shijiazhuang | SJZ | 114.529 | 38.028 | 70 | Urban |
| Shuangqing Road | SQL | 116.338 | 40.007 | 58 | Urban |
| Tanggu | TG | 117.717 | 39.044 | 13 | Urban |
| Tianjin | TJ | 117.206 | 39.075 | 2 | Urban |
| Tangshan | TS | 118.156 | 39.624 | 14 | Urban |
| Xianghe | XH | 116.962 | 39.754 | 9 | Suburban |
| Xinglong | XL | 117.576 | 40.394 | 879 | Rural |
| Yangfang | YF | 116.126 | 40.147 | 78 | Suburban |
| Yanjiao | YJ | 116.824 | 39.961 | 26 | Suburban |
| Zhangjiakou | ZJK | 114.918 | 40.771 | 777 | Urban |
| Zhuozhou | ZZ | 115.988 | 39.460 | 48 | Suburban |

Table 2. Comparisons of occurrences of exceedances of China's National Ambient Air Quality
Standards between observations and simulations[*]

| | Frequency | Class 1 | Class 2 | Obs. | 45-km | 15-km | 5-km |
|---|---|---|---|---|---|---|---|
| CO | Hourly | 10 | 10 | 1,150 | 0 | 0 | 0 |
| $O_3$ | Hourly | 160 | 200 | 3,684 | 24,807 | 10,283 | 9,880 |
| NOx | Hourly | 250 | 250 | 9,009 | 14 | 520 | 3,003 |
| $SO_2$ | Hourly | 150 | 500 | 393 | 0 | 2 | 39 |
| PM2.5 | 24-hours | 35 | 75 | 1,343 | 1,720 | 1,610 | 1,574 |
| PM10 | 24-hours | 50 | 150 | 2,834 | 2,067 | 1,617 | 1,676 |

* Class 1/2 standards are for rural/suburban-urban, respectively. Units are $\mu g\ m^{-3}$.




Table 3. Regional total emissions and average meteorology and air quality at various resolutions

| Variables | Period | 45-km | 15-km | 5-km |
|---|---|---|---|---|
| Biogenic Isoprene (tons) | Annual | 740,562 | 869,317 | 862,199 |
| Dust (tons) | Annual | 2,431 | 4,485 | 3,910 |
| Sea salt (tons) | Annual | 548 | 1,287 | 1,417 |
| Surface air temperature | January | 268 | 267 | 268 |
| (K) | July | 300 | 299 | 299 |
| Surface wind speed | January | 2.92 | 2.73 | 2.51 |
| ($m\ s^{-1}$) | July | 1.70 | 1.54 | 1.52 |
| SWDOWN | January | 124 | 117 | 117 |
| ($W\ m^{-2}$) | July | 249 | 242 | 250 |
| PBLH | January | 333 | 338 | 331 |
| (m) | July | 627 | 595 | 574 |
| CWP | January | 4.34 | 11.3 | 11.1 |
| ($g\ m^{-2}$) | July | 41.4 | 56.8 | 55.2 |
| Surface $O_3$ | January | 37.5 | 39.4 | 39.5 |
| (ppbv) | July | 86.8 | 68.8 | 69.2 |
| Surface NOx | January | 19.8 | 14.9 | 15.0 |
| (ppbv) | July | 9.03 | 8.32 | 7.96 |
| Surface CO | January | 0.600 | 0.521 | 0.526 |
| (ppmv) | July | 0.444 | 0.336 | 0.308 |
| Surface $SO_2$ | January | 16.6 | 12.9 | 13.2 |
| (ppbv) | July | 10.2 | 6.55 | 6.23 |
| Surface PM2.5 | January | 70.9 | 59.8 | 61.0 |
| ($\mu g\ m^{-3}$) | July | 89.3 | 58.0 | 46.2 |
| Surface PM10 | January | 102 | 88.1 | 89.6 |
| ($\mu g\ m^{-3}$) | July | 108 | 74.9 | 60.3 |







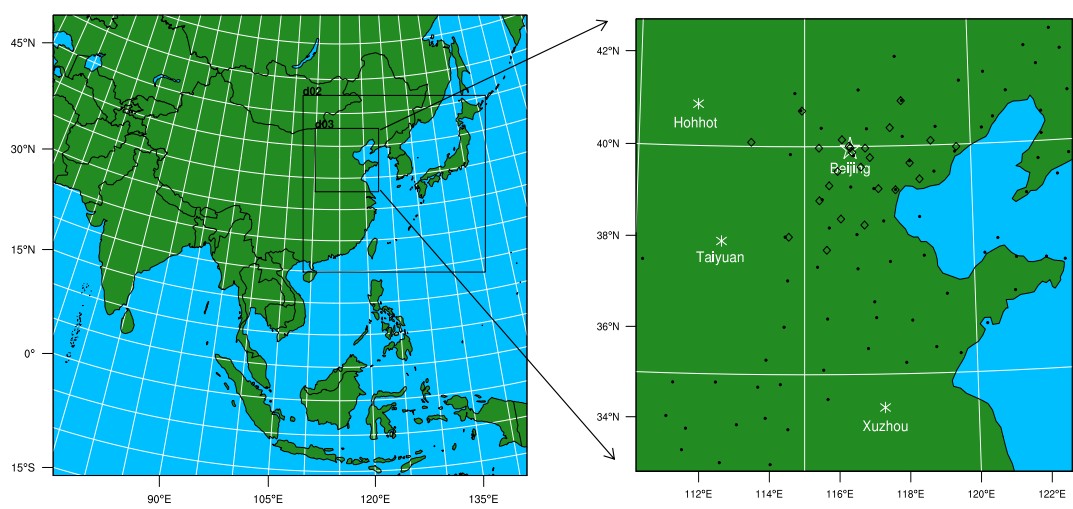

Figure 1. NU-WRF domain set-up. Left panel is the nested MICS-Asia domains; right panel is the
enlarged NCP domain (d03) with diamonds representing the air quality monitoring sites and black
dots denoting for the meteorological stations. Locations of four cities are marked for position
reference.






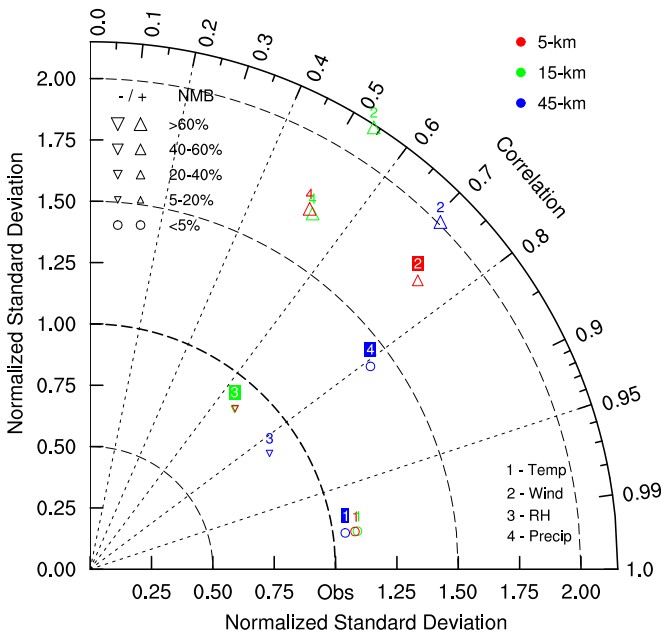


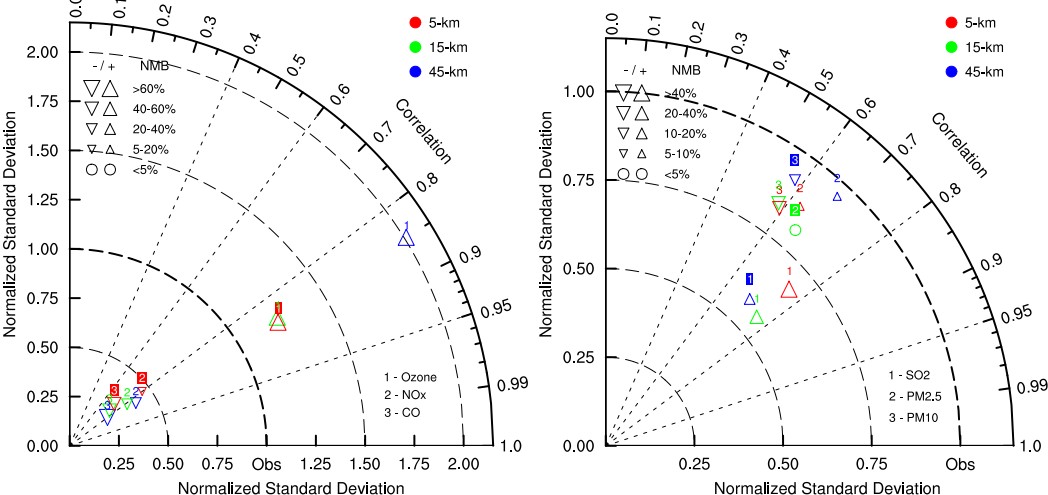

Figure 2. Taylor diagram for evaluations of NU-WRF performances on meteorology (top row) and
air quality (bottom row) simulations at three resolutions






Figure 3. Comparisons of *MB*, *RMSE*, and correlation coefficient ($r$) of surface O$_3$ from different
horizontal resolutions at each air quality monitoring site




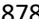

Figure 4. Comparisons of *MB*, *RMSE*, and correlation coefficient ($r$) of surface PM2.5 from different horizontal resolutions at each air quality monitoring site (blank space indicates no data are available)







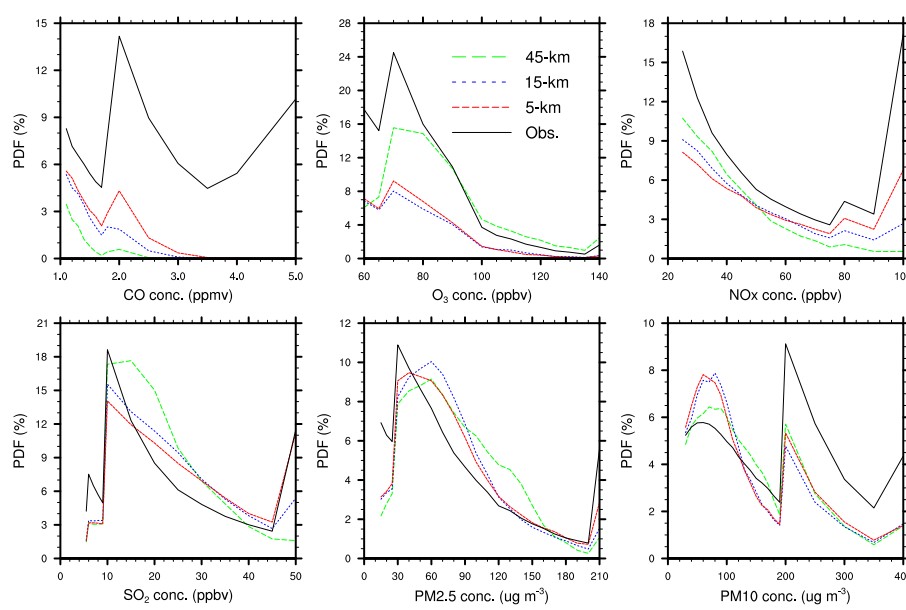

Figure 5. Probability density function (PDF) plots for hourly concentrations of surface air quality

Figure 6. Simulated emissions and July average meteorology from 3 grids: 1st row = isoprene
emissions (mol km-2 hr-1) from biogenic sources on a typical summer day; 2nd row = surface wind
vector with the shade representing wind speed (m s-1); 3rd row = surface air temperature (K); 4th
row = PBLH (m); 5th row = SWDOWN (W m-2); 6th row = CWP (g m-2).



Figure 7. Simulated January (SO₂, PM2.5, and PM10) and July (O₃, NOx, and CO) surface average
air quality from 3 grids: 1$^{st}$ row = O₃ (ppbv); 2$^{nd}$ row = NOx (ppbv) 3$^{rd}$ row = CO (ppmv); 4$^{th}$ row
= SO₂ (ppbv); 5$^{th}$ row = PM2.5 (μg m$^{-3}$); 6$^{th}$ row = PM10 ((μg m$^{-3}$).



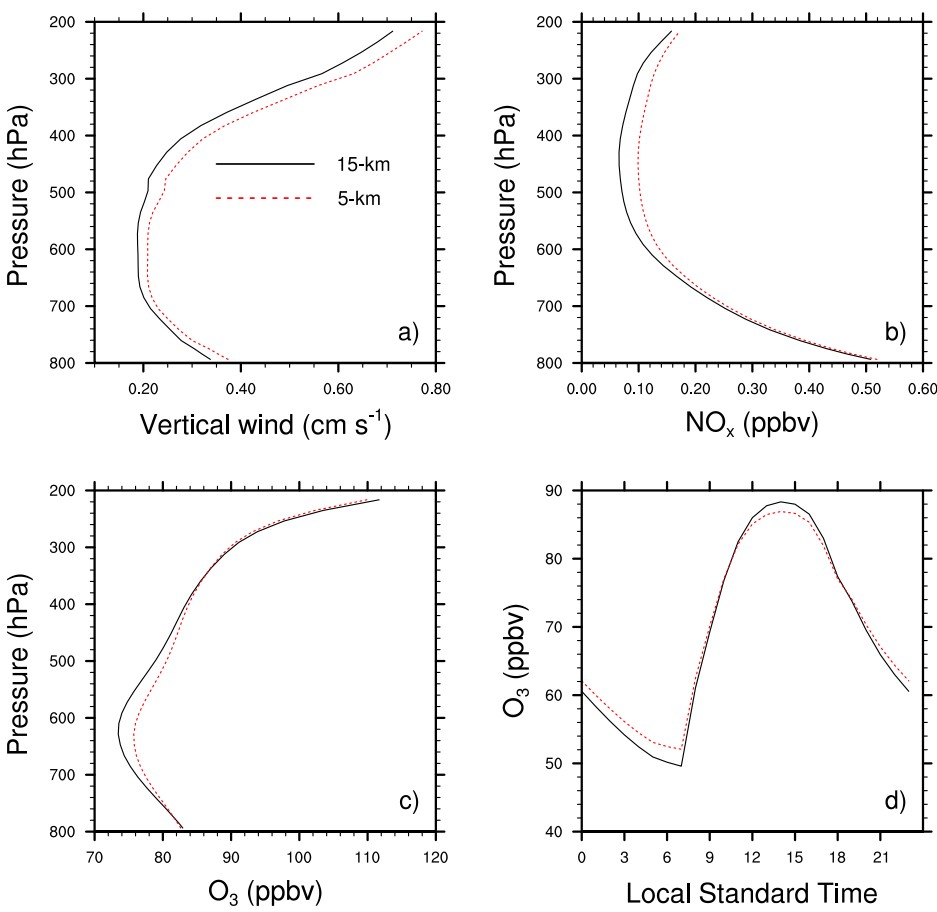

Figure 8. Domain average profiles of vertical wind, NOx, and O3 concentrations (Panels a~c) and
domain average diurnal variations of surface O$_3$ over July.