# Peer review of "Evaluation of NU-WRF Performance on Air Quality Simulation under Various Model Resolutions – An Investigation within Framework of MICS-Asia Phase III"

_Atmospheric Chemistry and Physics, 2019_

## Referee Comment (RC1) · Anonymous Referee #1 · 15 Oct 2019

General comments: This is a model evaluation paper with a focus on impact of model grid-spacing on meteorological and air quality simulations at regional scale. In this manuscript, the authors present extensive evaluations and inter-resolution comparisons of a year-long meteorological and air quality simulations conducted with the NANA Unified WRF model over the three nested domains in 2010 to identify the model uncertainties. The study represents a great interest by providing more evidence to further improvement of model performance on air quality prediction over heavily polluted regions such as North China Plain. The authors argue that no single resolution

can yield the best performance for all the variables across all the simulation and improvement of air quality prediction is not linear with increase in grid-spacing. Overall the paper is well written. However, some conclusions such as the recommendation of 15-km resolution may not be true in some important cases. More detailed information about model configuration and additional analyses are required to better quantify the impact of grid-spacing on both meteorological and air quality simulations. A major revision is suggested with a few reasons.

Major comments 1. The simulations conducted over the three nested domains are used in this study for comparisons and evaluations of meteorological and air quality simulations among three different grid-spacings. The way of nesting used in the simulations may have a large impact on the sensitivity of simulation results to grid spacing. Which nesting way is used in the simulations? One-way nesting or two-way nesting? If two-way nesting is selected in the simulations which I doubt according to results presented in the manuscript (e.g., L175-176), how your conclusion or findings can be affected by this selection?

2. The impact of grid-spacing on met or air quality simulations is highly dependent on terrain complexity of the study region. Given the relatively flat terrain of North China Plain (NCP), the impact of the grid-spacing on the simulations over land cannot be as large as expected. We believe that complex terrain and the situations with a large surface-cover contrast such as coastal regions do require high-resolution simulations. 15-km horizontal resolution that the authors suggest for the MICS-Asia study is definitely not enough to resolve the detailed local wind structures such as land-sea breeze or lake-breeze over the coastal regions which many large cities (e.g., Shanghai and Hong Kong) are located and air pollution is a big concern. It would be very helpful if the authors can shed little bit light on discussion of model performance with three grid-spacings at those sites along or near coastal regions.

3. Taylor diagram (Figure 2) is a useful way to present model performance, but it is not enough to represent model performance over a large region such as NCP and long-

time simulation period such as one year since model performs differently in different sub-regions like urban or rural areas and at time periods (e.g., different reasons). It will be helpful if the authors can provide any model performance in terms of spatial pattern (e.g., prediction biases) or time series of observation-simulation comparison. The result can be added in an appendix part if pages are limited.

4. Figure 7: It seems that simulated O3 spatial patterns are not matched well with that of its precursors including NOx simulations and isoprene emissions (see Fig.6) at different grid-spacing. For instance, the simulated surface NOx concentrations at the grid-spacings of 15-km and 5-km grids look very similar to those at the grid-spacing of 45-km. However, the simulated O3 concentrations out of the 15-km and 5-km grids are much smaller than those at the grid-spacing of 45-km. More explanations will be helpful to readers for better understanding their relationship and the model performance at varying grid-spacings.

5. L430-432: How the maximum PBLH can be observed in Mongolian Plain where surface cover is dominated by grass?

6. Table 3: Is it possible to add any available observational data for a comparison? The values presented in Table 3 represent domain average. It is not clear whether the simulations at those grids over ocean were included in the calculations.

Minor comments 1. L60: Is "CHIMERE" defined? Please check similar issue for other abbreviation terms. 2. L120: Is "off" correct? 3. L208: "simulated the best precipitation" or "simulated the precipitation best"? I recommend the latter. Please check the similar issue in several other places.

---

## Referee Comment (RC2) · Anonymous Referee #2 · 9 Nov 2019

Review

The manuscript presents a detailed investigation of NU-WRF at three different resolutions (45 km, 15 km, and 5 km) in simulating meteorology and air pollution over the North China Plain. Comparing model performance at different resolutions provides insights into model processes that are resolution dependent. The manuscript concludes that the 15-km resolution model has the overall best performance among the three. This is somewhat surprising as the finest resolution model is often assumed to be better, but the discussion provided in the manuscript is not sufficient to provide a

process-level understanding of why a particular resolution of the model outperforms, which is one major concern for the manuscript. Overall, the manuscript is well-written and will be suitable for ACP after the following comments are addressed.

Major comments: 1. For the most part, the manuscript provides only domain-mean comparison between the three resolutions against observations. Although site-level model evaluation is shown in the figures, they are mere statistics and lack follow-up investigations or discussions that can be linked to certain model processes or input data that can provide insights for model improvement or can be generalized for other regions and time periods. For example, more analysis should be conducted to examine where/when the variations in meteorology and air quality are the largest within the domain that are most challenging for the 5-km model to capture.

2. It is not clear whether the model input data are resolution aware. Are the underlying emissions inventory data and land surface data (topography, LAI, etc) at a fine resolution of 5 km and then aggregated to the coarser resolutions? If the model is not driven by inputs that can resolve 5-km surface conditions, the 5-km model will not be able to correctly simulate air pollution variations at the 5-km scale.

3. Figure 7, top panel: Ozone simulated by the 45-km model is almost 20 ppbv higher than the other two resolutions for July throughout the whole domain, while emissions of ozone precursors and meteorology are not so different. Why? Is this some kind of model error? If the model's oxidant budget is strongly resolution-dependent, one will question whether the model processes are parameterized correctly. A stable model should produce regional-mean concentrations of key species that are more or less consistent between different resolutions; it is the sub-regional variability and extreme concentrations that will differ as the resolution changes. This is reflected in ozone simulated by the 15-km and 5-km grids, but the 45-km model is an outlier.

4. Table 3: Natural emissions (isoprene, dust, and sea salt) are very different between the three resolutions, varying by almost a factor of two. While these emissions are dependent on meteorology and thus on the model resolution, the standard practice is to implement a scaling factor so that the domain-wide emissions are consistent between different resolutions. Otherwise, it will not be a fair comparison as the emissions are not constant across the three resolutions. As this manuscript is part of a model intercomparison study, these emissions should be consistent with other models participating in the study.

Minor comments: Line 215-210: the different conclusion from Gao et al. was due to the difference in observations or in the model setting?

Table 2: I don't understand this table. What are the numbers in each cell and why they are so different?

Line 32: add "the" before 21st century

Line 68: remove "however"

––––––––––––––––––––––––––––

---

## Author Comment (AC1) · 11 Dec 2019

Please to refer to the Authors' Comments (AC) for detail responses.

---

## Author Comment (AC2) · 11 Dec 2019

Please to refer to the Authors' Comments (AC) for detail responses.
* * *

---

## Author Comment (AC3) · 11 Dec 2019

**Evaluation of NU-WRF Performance on Air Quality Simulation under Various Model Resolutions –
An Investigation within Framework of MICS-Asia Phase III
By Z. Tao et al.**

The authors greatly appreciate the reviewers' insightful and constructive comments. We have made substantial revisions in an effort to address the key comments. The following lists our responses (plain text) to the reviewer's comments (bold).

*Response to Reviewer #1*

**1. The simulations conducted over the three nested domains are used in this study for comparisons and evaluations of meteorological and air quality simulations among three different grid-spacings. The way of nesting used in the simulations may have a large impact on the sensitivity of simulation results to grid spacing. Which nesting way is used in the simulations? One-way nesting or two-way nesting? If two-way nesting is selected in the simulations which I doubt according to results presented in the manuscript (e.g., L175-176), how your conclusion or findings can be affected by this selection?**

In this study, we chose the one-way nesting approach. If a two-way nesting approach is used, the meteorological fields for the study domain (innermost domain) from 3 horizontal resolutions would be very similar to each other, which would obscure the conclusion of the effect of model resolution on air quality. In the one-way nesting approach, the meteorological fields of the mother domains are independent on those of the respective nested domains, thus representing a cleaner signal of the grid resolution effect on air quality than that of the two-way nesting approach. As illustrated in Figure 6 of the manuscript, surface wind, temperature, PBLH, ground-level short wave radiation, and cloud water path out of each grid display noticeable differences. Due to the resource constrain, this investigation chose the one-way nesting approach, in which the lateral boundary conditions (LBCs) for each grid modeling were not identical. A "cleaner" approach would be to apply the identical LBCs out of the global model to drive NU-WRF under various grid resolutions to evaluate their impacts on air quality. However, we do not anticipate that such "cleaner" approach would remarkably alter the results based on the one-way nesting approach. In the manuscript, we added a statement in section 2 that the one-way nesting was utilized.

"…A one-way nesting approach was applied so that the values of the mother domains were independent on those of the respective nested domains…"

**2. The impact of grid-spacing on met or air quality simulations is highly dependent on terrain complexity of the study region. Given the relatively flat terrain of North China Plain (NCP), the impact of the grid-spacing on the simulations over land cannot be as large as expected. We believe that complex terrain and the situations with a large surface-cover contrast such as coastal regions do require high-resolution simulations. 15-km horizontal resolution that the authors suggest for the MICS-Asia study is definitely not enough to resolve the detailed local wind structures such as land-sea breeze or lake-breeze over the coastal regions which many large cities (e.g., Shanghai and Hong Kong) are located and air pollution is a big concern. It would be very helpful if the authors can shed little bit**

**light on discussion of model performance with three grid-spacings at those sites along or near coastal regions.**

In this investigation, we have found that the 5-km resolution modeling provided the best results of wind and surface pollutant levels, especially in polluted conditions that were the most relevant to air quality regulation (e.g., compliance of national air quality standards), measured by bias and RMSE. However, the improvement of model performance on air quality was not linear with the resolution increase. For example, the accuracy of modeled surface $O_3$ out of the 15-km grid was greatly improved over the one from the 45-km grid. Further increase of grid resolution to 5-km, however, showed diminished impact on model performance improvement on $O_3$ prediction for the study region. In addition, the cost in terms of cpu hours and disk space usage increased dramatically when adopting the 5-km grid, which would be a big hurdle for the inter-model comparison studies such as MICS-Asia that relied on community contributions to model Asia air quality over a relatively long time period. Considering all these factors, we suggest a 15-km resolution grid for future MICS-Asia modeling activity to achieve both accuracy and efficiency.

Of course, the choice of grid resolution also depends on the problems to be solved, such as air quality over coastal areas which show sharp contrasts of surface roughness, albedo, and thermal characteristics. In this investigation, QHD site locates approximately 5 km from the ocean and is subject to sea breeze effects. The temporal profiles of surface wind speed and temperature from the observation and model results out of 3 grids for QHD are shown in the following figure. The results indicated that the choice of grid resolution had large impacts on model simulations at this coastal site. The selection of the 5-km grid reduced biases of both surface temperature and wind speed. The biases of temperature reduced from 1.22 K (45-km) to -0.42 K (15-km), and further down to -0.31 K when the 5-km grid was applied. The biases of surface wind speed for the 45-km, 15-km, and 5-km grids were 3.72, 4.19, and 1.95 m s$^{-1}$, respectively. Since there were no hourly wind data available to this study, the diurnal changes of sea breeze cannot be evaluated. However, the benefit of finer resolution grid to improving wind simulation was obvious.

[Figure]

The following figure displays the time evolution of surface ozone and NOx concentrations from the observation and model results out of 3 grids for QHD. It can be seen that overall the model, regardless which grid resolution was applied, underestimated ground-level NOx concentrations but overestimated surface ozone levels. The ozone overestimate was especially large during summer months when its photochemical formation was the most efficient. We believe that the inaccurate NOx emissions representations were largely responsible for the model-observation mismatch. On the other hand, the benefit of increasing grid resolution to improving ozone and NOx forecast skills was obvious. The biases of ozone/NOx for the 45-km, 15-km, and 5-km resolution grids were 29.94/-22.46 ppbv, 24.09/-20.29 ppbv, 23.97/-17.95 ppbv, respectively. The respective RMSE were 37.24/28.87 ppbv, 27.28/27.57 ppbv, 27.01/26.38 ppbv. The improvement using the 15-km grid over the 45-km grid was remarkable but that using the 5-km grid over the 15-km grid was marginal.

[Figure]

In summary, the authors agree that, in general, the higher the grid resolution is, the better the simulation results will be. High resolution modeling is especially important to coastal areas and complex terrains where land-surface driving forces are in sharp contrast, such as QHD site. On the other hand, this research also agrees with the findings reported in many other papers that the benefit of higher resolution modeling of air quality starts to diminish at certain point due to the nonlinear nature of the atmospheric system. Balancing the modeling accuracy and computing resource constrain, a 15-km resolution grid has been recommended for future MICS-Asia activities if the investigate domain remains unchanged. We modified the manuscript to make this point explicitly stated in the section 3.1.2.b (Individual site) section 4 (Summary).

In section 3.1.2.b:

[revised manuscript text omitted]

**3. Taylor diagram (Figure 2) is a useful way to present model performance, but it is not enough to represent model performance over a large region such as NCP and long- time simulation period such as one year since model performs differently in different sub-regions like urban or rural areas and at time periods (e.g., different reasons). It will be helpful if the authors can provide any model**

**performance in terms of spatial pattern (e.g., prediction biases) or time series of observation-simulation comparison. The result can be added in an appendix part if pages are limited.**

Thanks for the suggestion. We have already had the statistics and discussions of each individual air quality site shown in Figures 3/4 and section 3.1.2.b. We added the time series of observation-simulation comparison averaged over the areas where the monitoring sites were located in the supplement material as shown in the following figures. We also inserted some discussions in section 3.1.1 for meteorological comparisons and in 3.1.2.a for regional average air quality comparisons.

[Figure]

Figure 1s. Time series of surface meteorological parameters of observed vs. modeled values at different grid resolutions in areas where the monitoring sites are located.

At the end of section 3.1.1

"The time series of daily mean wind speed, air temperature, and RH, as well as daily total precipitation averaged over the monitoring sites is illustrated in Figure 1s in the supplement material. It echoed the above findings based on the Taylor diagram. It appeared that NU-WRF constantly overestimated surface wind speed throughout the year with large overestimate occurring in fall and winter, while it severely underestimated RH in summer. Uncertainty in representation of land surface characteristics at least partially explained these biases (Yu, 2014; Gao et al., 2018). High-resolution grid tended to reduce the uncertainty in land surface representation, which would be helpful to improving model performance in meteorology simulation. A more detailed exploration of model-observation mismatch was insightful but beyond the scope of this research."

[Figure]

Figure 2s. Time series of surface air quality of observed vs. modeled values at different grid resolutions in areas where the monitoring sites are located.

At the end of section 3.1.2.a

"Figure 2s in the supplement material shows the time series of daily mean air quality averaged over the monitoring sites for the year 2010. The constant underestimate of CO throughout the year, severe underestimate of NOx in fall and winter, and large underestimate of SO₂ in summer all pointed out that the

emissions inventory may be incomplete, agreeing with the reports by Kong et al. (2019) and Li et al. (2019). In the future, improvement of the emissions inventory accuracy and more realistic temporal emissions distribution may help improving NU-WRF performance in simulating O₃ photochemistry."

**4. Figure 7: It seems that simulated O3 spatial patterns are not matched well with that of its precursors including NOx simulations and isoprene emissions (see Fig.6) at different grid-spacing. For instance, the simulated surface NOx concentrations at the grid-spacings of 15-km and 5-km grids look very similar to those at the grid-spacing of 45-km. However, the simulated O3 concentrations out of the 15-km and 5-km grids are much smaller than those at the grid-spacing of 45-km. More explanations will be helpful to readers for better understanding their relationship and the model performance at varying grid-spacings.**

This is a good point. Actually, the other reviewer also raised the similar question. The authors believe, through carefully analysis, that the following two factors play major roles in these results. 1) Ozone photochemistry: ozone is a secondary pollutant formed in the atmosphere in the presence of its precursors such as NOx and VOCs, as well as solar radiation. Except for limited urban areas, ozone formation is typically limited by the availability of NOx in the vast rural areas as illustrated in Figure 7. In this case, the 45-km grid tended to distribute NOx emissions more evenly in the region, effectively decreasing the surface NOx concentration in urban areas but increasing it over rural areas. The larger average wind speeds out of the 45-km grid (Figure 6 and Table 3) in July further smoothed out NOx distributions in NCP. This in turn increased the domain average surface O₃ concentration via photochemistry based on the 45-km resolution results. 2) Vertical lifting effect: fine resolution (e.g., 15-km and 5-km) modeling tended to produce stronger updraft than a coarse resolution modeling (e.g., 45-km) as shown in Figure 4s. This finding is consistent with the work by Lee et al. (2018) who account this partly for the aerosol-cloud interaction induced freezing/evaporation-related invigoration mechanism. The strong uplift would bring more surface pollutants such as NOx into the upper atmosphere, thus further reducing the NOx availability at ground that limits the surface ozone production but increases its formation in the upper atmosphere (see Figure 8 in the manuscript). In future studies, the measured vertical meteorology and pollutant profiles will be extremely helpful in elucidating the reasons.

[Figure]

Figure 4s. Domain average vertical wind speed at different grid resolutions along the altitude

A few sentences were added in section 3.2.3:
"…The domain average discussed in this section, however, was the average covering the vast rural area that generally was NOx-limited such that surface $O_3$ formation was controlled by the availability of NOx – more NOx resulting in more $O_3$ through photochemical processes. In this case, the 45-km grid tended to distribute NOx emissions more evenly in the region, effectively decreasing the surface NOx concentration in urban areas but increasing it over rural areas. The larger average July wind speed simulated by the 45-km grid (Figure 6 and Table 3) further smoothed out the NOx distribution in NCP. This in turn increased the domain average surface $O_3$ concentration via photochemistry based on the 45-km resolution results. In addition, vertical lifting played an important role in explaining the maximum regional $O_3$ in July simulated by the 45-km grid as compared to the results by the other two grid resolutions. As displayed in Figure 4s in the supplement material, a fine resolution modeling (e.g., 5-km) tended to produce a stronger updraft than a coarse resolution modeling (e.g., 45-km), consistent with the findings by Lee et al. (2018). The strong uplift would bring more surface pollutants such as NOx into the upper atmosphere, thus further reducing the NOx availability at ground limiting the surface ozone production but increasing its formation in the upper atmosphere."

**5. L430-432: How the maximum PBLH can be observed in Mongolian Plain where surface cover is dominated by grass?**

PBL growth is primarily driven by the buoyancy due to surface heating. Thus, PBLH is closely related to the sensible heating at surface. The larger the sensible heating is, the deeper the PBL will be (e.g., Tao et al., 2013). Meanwhile, the high sensible heating is generally associated with a dry soil as reported in Bindlish et al. (2001). Major vegetation coverages over the study domain include grasslands mosaiced with open shrublands (over large portions of the northwest quartile of the domain), croplands (over large portions of eastern part of the domain outside of water), various deciduous forests (areas separate grassland and cropland), and urban. The grassland soil is generally drier than that of other vegetation covers in the domain. This explains why the largest average PBLH is found over the grassland in the northwestern corner of the domain. The text of L432-434 has been modified as:

"…The large average PBLH (more than 1,000 m) was found in the northwestern corner of the domain with a dominant land cover type of grassland mosaiced with open shrubland that appeared to be drier than the other land cover types in the domain. The high sensible heating associated with dry soil tended to produce the deep PBL (Tao et al., 2013)."

**6. Table 3: Is it possible to add any available observational data for a comparison? The values presented in Table 3 represent domain average. It is not clear whether the simulations at those grids over ocean were included in the calculations.**

The purpose of Table 3 is to facilitate the analysis of inter-resolution model comparison (section 3.2). Therefore, no observational data is listed in this table. The comparisons with the observations have been presented in section 3.1. The regional averages presented in Table 3 were calculated including every grid

(land and ocean) within the domain. We changed the Table title to "Domain total emissions and average meteorology and air quality at various resolutions".

**7. L60: Is "CHIMERE" defined? Please check similar issue for other abbreviation terms.**

CHIMERE is not an abbreviation. It is the name of a Eulerian off-line chemistry-transport model developed in France. We modified the sentence (L61) as "…using the CHIMERE chemistry-transport model at various horizontal resolutions over Paris". We also checked the text and spelled out the abbreviation when it first occurred.

**8. L120: Is "off" correct?**

We modified the sentence to avoid confusion. The new description is "…new Grell cumulus scheme developed from the ensemble cumulus scheme that allowed subsidence spreading.".

**9. L208: "simulated the best precipitation" or "simulated the precipitation best"? I recommend the latter.**

We changed the sentence as suggested. We also checked the text to make the recommended changes as appropriate.

*Response to Reviewer #2*

**1. The manuscript concludes that the 15-km resolution model has the overall best performance among the three. This is somewhat surprising as the finest resolution model is often assumed to be better.**

In this investigation, we have found that the 5-km resolution modeling provided the best results of wind and surface pollutant levels, especially in polluted conditions that were more relevant to air quality regulation (e.g., compliance of national air quality standards), measured by bias and RMSE. However, the improvement of model performance on air quality was not linear with the resolution increase. For example, the accuracy of modeled surface $O_3$ out of the 15-km grid was greatly improved over the one from the 45-km grid. Further increase of grid resolution to 5-km, however, showed diminished impact on model performance on $O_3$ prediction for the study region. In addition, the cost in terms of cpu hours and disk space usage increased dramatically when adopting the 5-km grid resolution, which would be a big hurdle for the inter-model comparison studies such as MICS-Asia that relied on community contributions to model Asian air quality over a relatively long time period. Considering all these factors, we suggest a 15-km resolution grid for future MICS-Asia modeling activity to achieve both accuracy and efficiency. We checked the wording of the manuscript to make it clear that 15-km grid did not provide the best performance but rather was an optimal resolution that balanced the model accuracy and resource usages. For example, we modified the section 4 (Summary) as:

"…With regard to MICS-Asia Phase III whose major goal was to examine regional air quality, in general, the finer the grid resolution was, the better the simulation results would be. This was especially true over the coastal areas and complex terrains where a sharp local energy gradient existed. Fine resolution grid was also extremely helpful to reproducing pollutants at higher concentrations that were most relevant to air quality planning and management. However, the benefit of high resolution was not linear with the decrease of grid size. At certain point, the improved modeling accuracy due to an increase in grid resolution was so marginal that it cannot justify the computational cost associated with the fine grid simulation. Based on the balance of modeling accuracy and efficiency, a 15-km horizontal grid appeared to be an appropriate choice to optimize model performance and resource usage if the study domain remained unchanged for future MICS-Asia activities. The study suggested that the high-resolution emissions, especially the proper representation of emission gradients, would be helpful in improving air quality prediction. Moreover, the profile measurements of both meteorology and air quality, in supplement with the ground monitoring networks, would be greatly helpful to identifying model deficiencies and thus improving model forecast skills."

**2. For the most part, the manuscript provides only domain-mean comparison between the three resolutions against observations. Although site-level model evaluation is shown in the figures, they are mere statistics and lack follow-up investigations or discussions that can be linked to certain model processes or input data that can provide insights for model improvement or can be generalized for other regions and time periods. For example, more analysis should be conducted to examine where/when the variations in meteorology and air quality are the largest within the domain that are most challenging for the 5-km model to capture.**

This is a very good suggestion. We went back to data and made more analysis. Based on the results, we believe that the following factors account at least partially for the discrepancy between the modeled and observed air quality. 1) Spatial distribution of emissions was one key to determining air quality forecast accuracy. Out of 25 air quality monitoring sites used for model evaluation, 3 were rural sites and the remaining were urban/suburban sites. Figure 3s shows the typical time evolutions of surface ozone and NOx over the rural (XL) and urban (QHD) sites. It can readily be seen that NOx was underestimated at the urban site but overestimated at the rural site. The coarser the grid resolution, the severer the underestimates/overestimates were. This indicated that the 45-km resolution tended to smooth out emissions to make urban (or emissions centers) less polluted but rural more polluted. It in turn led to an overestimate of surface ozone over the urban sites mainly due to the reduced NOx titration effect, especially at night when there was no photochemical ozone formation. The statistics showed that the bias of the modeled daytime (7 am ~ 7 pm local time) average surface $O_3$ was 30% ~ 90% smaller than that of the daily average in the urban sites, no matter which grid resolution was applied. This suggests that, in the future, the high-resolution emissions, especially proper representation of emission gradients, will be helpful in improving air quality prediction. This point will be revisited in addressing comment 3.

[Figure]

Figure 3s. Time series of surface O$_3$ and NOx concentrations over QHD (upper) and XL (lower) sites of the observed vs. modeled values at different grid resolutions.

2) The driving meteorology, especially wind, was important to accurately forecast air quality. Take QHD site as an example. QHD site locates approximately 5 km from the ocean and is subject to sea breeze effects. There is a meteorological monitoring station co-locating at QHD. The temporal profiles of surface wind speed and temperature from the observation and model results out of 3 grids for QHD are shown in the following figure. The results indicated that the choice of grid resolution had large impacts on model simulations at this coastal site. The selection of the 5-km grid reduced biases of both surface temperature and wind speed. The biases of temperature reduced from 1.22 K (45-km) to -0.42 K (15-km), and further down to -0.31 K when the 5-km grid was applied. The biases of surface wind speed for the 45-km, 15-km, and 5-km grids were 3.72, 4.19, and 1.95 m s$^{-1}$, respectively. The improvement of meteorology forecast helped reducing the biases of air quality modeling. The biases of ozone/NOx for the 45-km, 15-km, and 5-km resolution grids were 29.94/-22.46 ppbv, 24.09/-20.29 ppbv, 23.97/-17.95 ppbv, respectively. The improvement using the 15-km grid over the 45-km grid was remarkable but that using the 5-km grid over the 15-km grid was marginal. Vertical wind profile was another important factor to determine surface air quality as shown in the answer to Comment 4. This emphasizes the importance to measure vertical profiles of both meteorology and air quality in the future, which will help improve model skills.

[Figure]

3) Photochemistry mechanism also impacts the model performance. This has been shown in the companion papers by Li et al. (2019) and Kong et al. (2019).

In summary, the authors find out that a high-resolution emissions inventory would greatly help improving the model performances, especially over urban areas and emissions centers. Over the coastal areas (e.g., QHD) and complex terrain areas (e.g., XL), high resolution modeling tends to produce a more realistic wind field that benefits air quality simulation. In the future, the profile measurements of both meteorology and air quality are needed to elucidate the discrepancy between simulation and observation, thus help to improve model skills. We added discussions in section 3.1.2.b (Individual site), section 3.2.3 (see answer to Comment 4), and section 4 (see answer to Comment 1) to reflect the above analysis.

In section 3.1.2.b:

"An effort has been put to identify the potential reasons that caused the model-observation discrepancy. First and as discussed previously, the spatial distribution of emissions was one key to determining air quality forecast accuracy. Figure 3s shows the typical time evolutions of surface $O_3$ and NOx over the rural (XL) and urban (QHD) sites. It can readily be seen that NOx was underestimated at the urban site but overestimated at the rural site. The coarser the grid resolution, the severer the underestimates/overestimates were. This indicated that the 45-km resolution tended to smooth out emissions to make urban (or emissions centers) less polluted but rural more polluted. It in turn led to an overestimate of surface $O_3$ over the urban sites mainly due to the reduced NOx titration effect, especially at night when there was no photochemical $O_3$ formation. The statistics showed that the bias of the modeled daytime (7 am ~ 7 pm local time) average surface $O_3$ was 30% ~ 90% smaller than that of the daily average in the urban sites, no matter which grid resolution was applied. This suggested that in the future the high-resolution emissions, especially proper representation of emission gradients, would be helpful in improving air quality prediction. The effect of emissions gradients associated with the grid resolution would be further discussed in the inter-model comparison section.

Next, the driving meteorology, especially wind, was important to accurately forecast air quality over coastal areas that bore sharp thermal contrasts. QHD site locates approximately 5 km from the ocean and is subject to sea breeze effects. The detailed analysis of meteorology and air quality over QHD was conducted. The results indicated that the choice of grid resolution had large impacts on model simulations at this coastal site. The selection of the 5-km grid reduced biases of both surface temperature and wind speed. The biases of temperature reduced from 1.22 K (45-km) to -0.42 K (15-km), and further down to -0.31 K when the 5-km grid was applied. The biases of surface wind speed for the 45-km, 15-km, and 5-km grids were 3.72, 4.19, and 1.95 m s$^{-1}$, respectively. The improvement of meteorology forecast helped reducing the biases of air quality modeling. The biases of $O_3$/NOx for the 45-km, 15-km, and 5-km resolution grids were 29.94/-22.46 ppbv, 24.09/-20.29 ppbv, 23.97/-17.95 ppbv, respectively. The improvement using the 15-km grid over the 45-km grid was remarkable but that using the 5-km grid over the 15-km grid was marginal. The result emphasized the importance of high-resolution modeling to improvements of air quality forecast skills, especially at coastal and complex terrain areas (e.g., QHD and XL)."

**3. It is not clear whether the model input data are resolution aware. Are the underlying emissions inventory data and land surface data (topography, LAI, etc) at a fine resolution of 5 km and then aggregated to the coarser resolutions? If the model is not driven by inputs that can resolve 5-km surface conditions, the 5-km model will not be able to correctly simulate air pollution variations at the 5-km scale.**

Thanks for raising a very important point. In addition to the computation constrain, the challenge to employing an ultra-fine resolution modeling is the availability of the input data that are at the same or similar resolution. In this study, the land surface data were derived from the 30s resolution (around 30 m along mid-latitude) MODIS products that were aggregate to the model resolution. However, the MIX anthropogenic and GFEDv3 fire emissions inventories utilized in the study have a resolution of 0.25 by 0.25 degree and 0.5 by 0.5 degree, respectively. As indicated in the answer to Comment 2 above, the uncertainty of emissions may lead to air quality modeling errors. Therefore, the resolution-aware emissions may further improve the model performance using a 5-km grid. We added a caveat in section 4 (Summary) to reflect this.

"… It also was worth noting that the benefit of increasing grid resolution to better surface O₃ and PM2.5 simulations started to diminish when the horizontal resolution reached 15-km, agreeing with the finding by Valari and Menut (2008). There is a caveat, though. The anthropogenic MIX and fire GFEDv3 emissions inventories bore the 0.25º by 0.25º and 0.5º by 0.5º resolution, respectively. These resolutions cannot resolve the 5-km grid. Should a 5-km resolution emissions inventory be available and used, the benefit of high-resolution modeling would likely be more prominent.".

**4. Figure 7, top panel: Ozone simulated by the 45-km model is almost 20 ppbv higher than the other two resolutions for July throughout the whole domain, while emissions of ozone precursors and meteorology are not so different. Why? Is this some kind of model error? If the model's oxidant budget is strongly resolution-dependent, one will question whether the model processes are parameterized correctly. A stable model should produce regional-mean concentrations of key species that are more or less consistent between different resolutions; it is the sub-regional variability and extreme concentrations that will differ as the resolution changes. This is reflected in ozone simulated by the 15-km and 5-km grids, but the 45-km model is an outlier.**

Thanks for pointing this out. Actually, the first reviewer also raised the similar question. The authors believe, through carefully analysis, that the following two factors play major roles in these results. 1) Ozone photochemistry: ozone is a secondary pollutant formed in the atmospheric in the presence of its precursors such as NOx and VOCs, as well as solar radiation. Except for limited urban areas, ozone formation is typically limited by the availability of NOx in the vast rural areas as illustrated in Figure 7. In this case, the 45-km grid tended to distribute NOx emissions more evenly in the region, effectively decreasing the surface NOx concentration in urban areas but increasing it over rural areas. The larger average wind speeds out of the 45-km grid (Figure 6 and Table 3) in July further smoothed out NOx distributions in NCP. This in turn increased the domain average surface O₃ concentration via photochemistry based on the 45-km resolution results. Actually, the spatial distributions of annual average surface O₃ out of three grids appeared to be less variable. 2) Vertical lifting effect: fine resolution (e.g., 15-km and 5-km) modeling tended to produce a stronger updraft than a coarse resolution modeling (e.g., 45-km) as shown in Figure 4s. This finding is consistent with the work by Lee et al. (2018) who account this partly for the aerosol-cloud interaction induced freezing/evaporation-related invigoration mechanism. The strong uplift would bring more surface pollutants such as NOx into the upper atmosphere, thus further reducing the NOx availability at ground that limits the surface ozone production but increases its formation in the upper atmosphere (see Figure 8 in the manuscript). In future studies, the measured vertical meteorology and pollutant profiles will be extremely helpful in elucidating the reasons.

A few sentences were added in section 3.2.3:
"…The domain average discussed in this section, however, was the average covering the vast rural area that generally was NOx-limited such that surface O₃ formation was controlled by the availability of NOx – more NOx resulting in more O₃ through photochemical processes. In this case, the 45-km grid tended to distribute NOx emissions more evenly in the region, effectively decreasing the surface NOx concentration in urban areas but increasing it over rural areas. The larger average July wind speed simulated by the 45-km grid (Figure 6 and Table 3) further smoothed out the NOx distribution in NCP. This in turn increased the domain average surface O₃ concentration via photochemistry based on the 45-km resolution results. In addition, vertical lifting

played an important role in explaining the maximum regional $O_3$ in July simulated by the 45-km grid as compared to the results by the other two grid resolutions. As displayed in Figure 4s in the supplement material, a fine resolution modeling (e.g., 5-km) tended to produce a stronger updraft than a coarse resolution modeling (e.g., 45-km), consistent with the findings by Lee et al. (2018). The strong uplift would bring more surface pollutants such as NOx into the upper atmosphere, thus further reducing the NOx availability at ground limiting the surface ozone production but increasing its formation in the upper atmosphere."

[Figure]

Figure 4s. Domain average vertical wind speed at different grid resolutions along the altitude

**5. Table 3: Natural emissions (isoprene, dust, and sea salt) are very different between the three resolutions, varying by almost a factor of two. While these emissions are dependent on meteorology and thus on the model resolution, the standard practice is to implement a scaling factor so that the domain-wide emissions are consistent between different resolutions. Otherwise, it will not be a fair comparison as the emissions are not constant across the three resolutions. As this manuscript is part of a model intercomparison study, these emissions should be consistent with other models participating in the study.**

We treated the biogenic, dust, and seasalt emissions that were calculated online as part of the effect of grid resolutions on air quality since the meteorological driving forces of these emissions, such as temperature, solar radiation, and wind, were impacted by the choice of grid resolutions. We think this is a fair justification.

**6. Line 215-210: the different conclusion from Gao et al. was due to the difference in observations or in the model setting?**

Gao et al. investigated the grid resolution effect on precipitation over the contiguous U.S. Their domain, modeling setup, and observations were all different from the ones used in this study. More

importantly, Gao et al. used the processed precipitation data for their model evaluation – their precipitation data were based on the daily rain gauge data that were gridded to the 0.125° resolution using the synergraphic mapping algorithm with topographic adjustment to the monthly precipitation climatology. The processed data promoted the precipitation homogeneity and reduced the chances of model-observation mismatch of a precipitation event. This may be the major reason that two studies draw the opposite conclusions. In the manuscript we emphasized that our conclusion was based on the comparison with the site observation. For example, in section 4 (Summary):

"…The statement on precipitation should be taken with caution since it was based on the comparison with the site observations. Seeing the very heterogeneous nature of precipitation, the penalty of model hitting or missing a rain event was severe. Thus, the coarse grid covering more areas within a grid cell would reduce chances of mistaken precipitation hitting or missing simulations. However, a comparison of modeled precipitations to gridded "observation" that was re-constructed using the synergraphic mapping algorithm with topographic adjustment to the monthly precipitation climatology showed opposite result, where the fine resolution modeling showed superior reproduction of precipitation than the coarse resolution simulation (Gao et al., 2017)."

**7. Table 2: I don't understand this table. What are the numbers in each cell and why they are so different?**

Table 2 lists the occurrences of exceedances of China's National Ambient Air Quality Standards (NAAQS). Column "Frequency" indicates the time integration of each NAAQS. Column "Class 1" lists the NAAQS for rural sites, and "Class 2" lists the standards for urban-suburban sites. "Obs" lists the occurrences of NAAQS exceedances for each pollutant based on the observations. Columns "45-km", "15-km", and "5-km" list the occurrences of NAAQS exceedances based on the modeling results using "45-km", "15-km", and "5-km" grid resolutions. We added a sentence in section 3.1.2.c:

"Table 2 lists the occurrences of violations of China's national ambient air quality standards (NAAQS) for the six pollutants from both observations and simulations, in which columns "Class 1" and "Class 2" list the standards for rural and urban-suburban sites, respectively, and column "Frequency" indicates the time integration of each NAAQS."

**8. Line 32: add "the" before 21st century.**
Done.

**9. Line 68: remove "however".**
Done.

---

## Author Response (AR2)

**Evaluation of NU-WRF Performance on Air Quality Simulation under Various Model Resolutions –**
**An Investigation within Framework of MICS-Asia Phase III**
**By Z. Tao et al.**

The authors greatly appreciate the reviewer's recommendation of acceptance of the manuscript for publication. We have made the minor revision as suggested. The following lists our responses (plain text) to the reviewer's comments (bold).

**1. L6: add "simulations" after "meteorology and air quality".**

Done.

**2. L75: L140-141: What is the frequency that the aerosol LBCs are updated with the GOCART simulation?**

The aerosol LBCs from GOCART were updated every 6 hours. The sentence in L140-141 has been modified as:

"The aerosol LBCs were from the global GOCART simulation with a resolution of 1.25 (longitude) by 1 (latitude) degree (Chin et al., 2007) updated every 6 hours."

**3. Figures 2,3,4,5: Please specify the time periods that the model performance represents in the figure captions.**

Done.